# Exploring Determinants of Successful Weight Loss with the Use of a Smartphone Healthcare Application: Secondary Analysis of a Randomized Clinical Trial

**DOI:** 10.3390/nu16132108

**Published:** 2024-07-02

**Authors:** Yutong Shi, Yuki Sasaki, Keiko Ishimura, Shinichiro Mizuno, Yoshio Nakata

**Affiliations:** 1Graduate School of Comprehensive Human Sciences, University of Tsukuba, Tsukuba 305-8574, Ibaraki, Japan; sekitong0426@gmail.com; 2Wellmira Inc., Chiyoda-ku, Tokyo 101-0041, Japan; y.sasaki@wellmira.jp (Y.S.); s.mizuno@wellmira.jp (S.M.); 3Institute of Health and Sport Sciences, University of Tsukuba, Tsukuba 305-8574, Ibaraki, Japan

**Keywords:** smartphone application, successful weight loss, predictors, overweight, obesity

## Abstract

Dietary and physical activity interventions through smartphone healthcare applications (apps) have recently surged in popularity as effective methods for weight loss. However, the specific factors contributing to successful weight loss remain uncertain. We conducted an analysis of baseline characteristics and app usage frequencies over three months among 68 Japanese adults with overweight and obesity who were assigned to the intervention group in a previous randomized controlled trial. Logistic regression analysis revealed a negative association (OR: 0.248; *p* = 0.018) between having a walking habit at baseline and successful weight loss, defined as a 3% reduction in initial weight. Additionally, slower walking speeds and family medical history were identified as potential predictors of successful weight loss. These findings offer insights into the profile of individuals who achieve success in weight loss through our smartphone app, providing valuable guidance for the development of future healthcare apps.

## 1. Introduction

The global prevalence of overweight and obesity presents a significant public health challenge. With more than one billion individuals with obesity worldwide currently, this figure continues to rise [1]. Obesity is closely linked to numerous non-communicable diseases, including type 2 diabetes, cardiovascular disease, hypertension, stroke, various forms of cancer, and mental health issues [1]. Consequently, there is a pressing need to address excessive weight gain to mitigate the prevalence of overweight and obesity. Effective strategies for treating overweight and obesity are essential. According to the 2013 AHA/ACC/TOS guideline, conventional weight loss interventions incorporating lifestyle modifications have proven efficacy [2]. These interventions typically integrate dietary changes, physical activity (PA), and behavioral adjustments to facilitate weight loss or maintenance [2].

Recently, there has been a growing emphasis on mobile- or smartphone-based lifestyle interventions for weight loss. Utilizing fitness-related smartphone applications (apps) to promote behavioral changes has demonstrated sustainable increases in PA and holds significance in public health efforts. Smartphone healthcare apps have the capacity to overcome numerous barriers to large-scale interventions, providing users with dietary and behavioral guidance to monitor their lifestyles effectively [3]. Our systematic review underscored the effectiveness of web-based interventions for weight loss, particularly highlighting the efficacy of the provision of personalized information [4].

In our prior randomized controlled trial, the smartphone healthcare app, CALO mama Plus (https://calomama.com/, accessed on 28 June 2024, Wellmira Inc., former Link & Communication Inc., Tokyo, Japan), that is under investigation in the present study demonstrated effectiveness for weight loss [5]. The app allows users to easily record daily lifestyle information and offers instant advice based on this data, utilizing artificial intelligence (AI) for personalized feedback. Despite these capabilities, it remains unclear which individuals benefit most from using the app and which factors contribute most effectively to successful weight loss. Therefore, to enhance the app’s personalized features, it is imperative to examine the fundamental characteristics of successful weight loss achievers and their patterns of app utilization.

Previous studies have indicated that early behaviors, such as self-monitoring of diet or PA, strongly predict the effectiveness of weight loss interventions [6,7,8]. West et al. found that weight changes prior to treatment initiation may influence the effectiveness of obesity treatments [6]. Krukowski et al. emphasized the significance of an early focus on self-monitoring behaviors within weight loss programs [7]. Additionally, Unick et al. suggested a positive association between weight loss in the initial two months and long-term weight reduction [8]. Thus, this study seeks to identify predictive factors and characteristics of individuals who successfully shed weight using our smartphone app.

## 2. Materials and Methods

### 2.1. Study Design

This study entailed a secondary analysis of data from a previous three-month randomized controlled trial conducted between 1 December 2020 and 22 February 2021 [5]. The original trial investigated the impact of a smartphone healthcare app on weight loss among Japanese adults with overweight or obesity, comparing them to a control group that received no interventions. The ethics review board of the Faculty of Health and Sport Sciences at the University of Tsukuba approved the study protocol (approval number: Tai 019-127; 21 January 2020). Owing to the COVID-19 pandemic, we decided to conduct all the sessions online. The revised protocol was approved on 1 October 2020 and registered at the University Hospital Medical Information Network (UMIN) Clinical Trial Registry (UMIN000042072) on 10 October 2020.

### 2.2. Participants

As detailed in the previous study [5], participants were recruited from four cooperating companies. Following the COVID-19 pandemic, all sessions were transitioned to online platforms. The initial introductory session was conducted via Zoom (https://explore.zoom.us/ja/products/meetings/, accessed on 28 June 2024, San Jose, CA, USA) or YouTube (https://www.youtube.com/, accessed on 28 June 2024, San Bruno, CA, USA). Interested and potentially eligible participants underwent a screening process, which included an online baseline questionnaire and self-administered measurements. Written informed consent was obtained from all participants, who were offered a reward valued at JPY 20,000.

Inclusion criteria were as follows: (1) age 20–65 years, (2) body mass index 23–40 kg/m^2^, (3) ability to install the healthcare app on their smartphones, and (4) office workers with a clear understanding of the study’s purpose and content and who provided written informed consent. Exclusion criteria were as follows: (1) a history of heart or cerebrovascular disease, (2) current pregnancy or plans for pregnancy during the study period, (3) significant weight fluctuations in the past six months, (4) participation of cohabiting family members in the study, and (5) being deemed unsuitable for the study by the principal investigator for other reasons (e.g., high blood pressure, high glucose level).

Of the eligible participants (*n* = 141), 69 were randomized into the control group and 72 into the intervention group. Of the intervention group, one participant became too busy to continue the trial, resulting in unavailable data after three months; one used an older version of the healthcare app, rendering app usage data uncollectable; and two had missing values for basic characteristics. Hence, data from 68 participants in the intervention group were utilized for analysis in this study.

### 2.3. Interventions

All interventions were conducted online in response to the COVID-19 pandemic. The intervention group received access to the smartphone healthcare app. Additionally, participants in the control group were given the opportunity to utilize the app for three months following the trial period.

Participants in the intervention group were instructed to log their lifestyle information into the app and to adjust their daily routines based on the app’s recommendations. The app also supplied target values for energy intake, nutrients, and PA, which were automatically calculated from baseline data. Utilizing AI, the app facilitated accurate recording by analyzing users’ meal photos, converting input data into numerical values. Appropriate advice was generated by comparing these converted values with the target values.

### 2.4. Measurements

All outcomes were evaluated at baseline and after three months. Participants were instructed to record their weight, wear an accelerometer to track PA, and complete questionnaires. The primary outcome assessed was the amount of weight lost over the three-month period. Each participant was provided a weight scale (HD-665; Tanita, Tokyo, Japan) to measure their weight.

#### 2.4.1. Basic Characteristics and Body Weight

The participants’ basic characteristics were gathered using self-administered questionnaire software (https://questant.jp/, accessed on 28 June 2024, Qusetant, Macromill, Tokyo, Japan). Items included age, sex, height, weight, exercise habits, walking habits, walking speed, medical history, medication usage, family medical history, current smoking status, employment status, educational background, household income, and living arrangements. To evaluate exercise habits, walking habits, and walking speed, participants answered “yes” or “no” to the following questions: “Do you engage in light, sweaty exercise for at least 30 min at a time, at least twice a week, for at least one year?”; “Do you walk or engage in similar PA for at least 1 h per day in your daily life?”; and “Do you walk faster than your peers of about the same age?”. Each participant’s body weight was measured using a weight scale (HD-665; Tanita, Tokyo, Japan) provided to them; weight data were collected via Qusetant at baseline and after three months.

#### 2.4.2. Dietary Intake and PA

Data from the dietary logs recorded within the app by participants in the intervention group were utilized to calculate energy intake.

PA was assessed using a validated triaxial accelerometer (Active style Pro HJA-750C; Omron Healthcare, Kyoto, Japan), which was capable of counting steps and estimating PA intensity using a published algorithm [9,10]. A valid daily record was defined as a wearing time of at least 10 h per day, with records of less than three valid days excluded [11,12]. Additionally, the app collected step counts if participants consented to synchronize their data with their smartphones.

#### 2.4.3. App Usage

Participants were encouraged to log their daily diet (breakfast, lunch, dinner, and snacks), exercise, weight, mood, and sleep within the smartphone app. The frequency of inputs for each item served as an indicator of adherence to the intervention. Study staff monitored the frequency of data inputs and emailed participants if dietary inputs were fewer than four days per week to encourage consistent logging. The number of entries during the first week was selected as the baseline, as no reminders were provided by study staff during this period.

### 2.5. Statistical Analysis

All statistical analyses were conducted using SPSS software (version 29.0; IBM Corp., Armonk, NY, USA). Statistical significance was set at *p* < 0.05. Continuous variables were presented as the mean (standard deviation [SD]), while categorical variables were expressed as numbers (percentage). Spearman’s rank correlation coefficients were computed between app usage frequency and weight loss.

Basic characteristics obtained from the baseline questionnaire and app usage frequency during the first week were included in both univariable and multivariable logistic regression analyses to assess successful weight loss. Results were reported as odds ratios (ORs), 95% confidence intervals (CIs), and *p*-values. Successful weight loss was defined as a 3% reduction in initial body weight [13]. Variables demonstrating statistical significance in the univariate logistic regression analysis were incorporated into the multivariate logistic regression analysis to establish the final model.

## 3. Results

### 3.1. Baseline Characteristics and Change in Body Weight

Table 1 presents the self-reported baseline characteristics of the 68 participants. Among them, 11 (16.2%) experienced weight gain, 32 (47.1%) lost < 3%, 12 (17.6%) lost 3–5%, and 13 (19.1%) lost > 5% of their initial body weight. The mean weight loss after 3 months was 2.4 kg (SD: 4.1 kg).

### 3.2. App Usage Frequency

During the first week, participants logged their diet, exercise, weight, mood, and sleep an average of 3.21, 0.48, 0.82, 0.80, and 0.86 times per day, respectively (Table 2). Figure 1 illustrates the weekly frequency of each item over the three-month period.

### 3.3. Correlation of App Usage Frequency and Weight Loss

Figure 2 illustrates the correlation matrix between the number of app inputs per day during the first week and weight loss over three months. No significant correlations were identified between app input frequency and weight loss. The most notable positive correlation was observed between mood and sleep inputs during the first week (ρ = 0.78; *p* < 0.001).

### 3.4. Logistic Regression Analysis

To explore the factors potentially linked with a 3% loss in initial body weight, we constructed a regression model incorporating baseline participant characteristics and app usage frequencies.

In the univariate analysis (Table 3), a walking habit (OR: 0.308; *p* = 0.026) and faster walking speeds (OR: 0.258; *p* = 0.011) were negatively associated, while a family medical history (OR: 3.929; *p* = 0.049) was positively associated with successful weight loss. These variables were then included in the multivariate analysis, where having a walking habit remained a statistically significant determinant that was negatively associated with successful weight loss (OR: 0.248; *p* = 0.018).

In addition, the changes in the step counts of those who did not have a walking habit and those who did were analyzed. After three months, those who did not have a walking habit increased their step count by 281 steps/day, while those who had a walking habit decreased it by 1168 steps/day (*p* = 0.056).

## 4. Discussion

Our previous randomized controlled trial [5] established the efficacy of the smartphone healthcare app for facilitating weight loss over a three-month period. This secondary analysis sought to identify predictive factors and characteristics associated with successful weight reduction among users of our smartphone app. Baseline participant characteristics and app usage frequencies during the first week were considered as candidate variables. Successful weight loss was defined as achieving a 3% loss in initial weight [13]. Univariate and multivariate logistic regression analyses revealed that not having a walking habit was a significant determinant of successful weight loss, alongside slower walking speeds and a family medical history, which were identified as potential predictors.

We posited baseline characteristics and app usage frequencies in the first week as potential determinants of successful weight loss. Previous research suggests that individuals who achieve clinically significant weight loss tend to have a better BMI and PA levels, favoring exercise over diet as their weight loss strategy [14]. Moreover, studies have highlighted the positive impacts of early PA goal attainment and initial weight loss effects on long-term weight management [15,16,17]. Conversely, some studies have reported a decline in adherence to self-monitoring using smartphone apps over time [18,19,20]. However, in our study, as depicted in Figure 1, the change in the weekly input frequency for each item remained stable rather than decreasing, which is likely attributed to staff reminders for participants with low compliance. Given that reminders were initiated from the second week based on inputs from the previous week, we included data from the first week to ensure comparative fairness in compliance.

Our logistic regression analyses indicated that app usage variables were not significant predictors of successful weight loss over the three-month intervention period. However, the original randomized controlled trial showed that people with obesity who used the app lost significantly more weight than those who did not [5]. This study’s findings also contrast with a previous study utilizing the same app, which demonstrated a statistically significant association between app usage and weight loss [21]. This may have been caused by sampling bias in the randomized controlled trials. As the participants in the randomized controlled trial were highly motivated to lose weight, their input status was good during the first week, even without reminders from the research staff, as shown in Figure 1. Therefore, the app input during the first week it was used in the study may not be a significant predictor of successful weight loss.

Additionally, our analyses identified not having a walking habit and slower walking speeds as significant determinants of successful weight loss. Similar findings were observed in the Diabetes Prevention Program, where participants with lower baseline PA levels achieved greater weight loss [22]. This study revealed similar findings; that is, the post-intervention step count of those without a walking habit increased more than that of those with a walking habit (*p* = 0.056). Therefore, these variables could serve as proxies for predicting successful weight loss.

A previous systematic review on the effectiveness of web-based feedback interventions indicated that individuals who received personalized feedback were more likely to achieve successful weight loss compared to those who did not receive feedback [23]. The smartphone app utilized in our study automatically provides real-time step counts and offers tailored walking suggestions. For instance, prompts such as “You’ve been sedentary for a while, why not take a 5-min walk?” or “Your current step count is 4000, with 6000 more steps to reach 10,000. Why not go for a short walk?”. Additionally, as depicted in Figure 1, participants in our study exhibited high adherence to the app, likely due to the reminders from the research staff. Consequently, it can be inferred that individuals lacking a walking habit and walking at a slower pace tended to respond positively to app feedback, thereby becoming more mindful of their walking behavior and increasing their energy expenditure.

Having a family medical history also emerged as a potential determinant associated with successful weight loss. In a previous study, a weight loss intervention targeting first-degree relatives with type 2 diabetes proved significantly more effective than the control group [24]. Consequently, participants with a family medical history may possess heightened health awareness, leading to increased engagement in risk-reducing behaviors.

These findings offer insights into the characteristics of individuals who achieve successful weight loss using our smartphone app and can inform the development of future healthcare apps, which can also be utilized in clinical settings. However, this study is subject to some limitations. First, all participants were Japanese, and the sample size was small, with the smartphone app’s text being in Japanese. Thus, the generalizability of these results may be limited. Second, our results are based on a three-month trial, precluding adaptation to long-term weight loss or maintenance interventions. Third, the BMI criterion for participants included in this study was being over 23 kg/m^2^, which is different from the Japanese definition of people with obesity (BMI ≥ 25 kg/m^2^). Considering the low prevalence of obesity in Japan and the lower threshold of increasing risk, we designated a BMI of 23 kg/m^2^ to allow for recruitment for this study, which was explained in the original article [5]. Finally, the study staff checked data input frequencies to enhance the adherence of the intervention group in our study. However, this may not reflect adherence to the health app in the real world. Therefore, a replication in real-world situations needs to be considered for future research.

## 5. Conclusions

This secondary analysis of data from a previous randomized controlled trial identified not having a walking habit at baseline as a significant predictive factor for successful weight loss, while slower walking speeds and having a family medical history were deemed potential factors. These findings will inform the development of additional personalized functions within the studied app. Further research is warranted to extend these findings to other app usage contexts.

## Figures and Tables

**Figure 1 nutrients-16-02108-f001:**
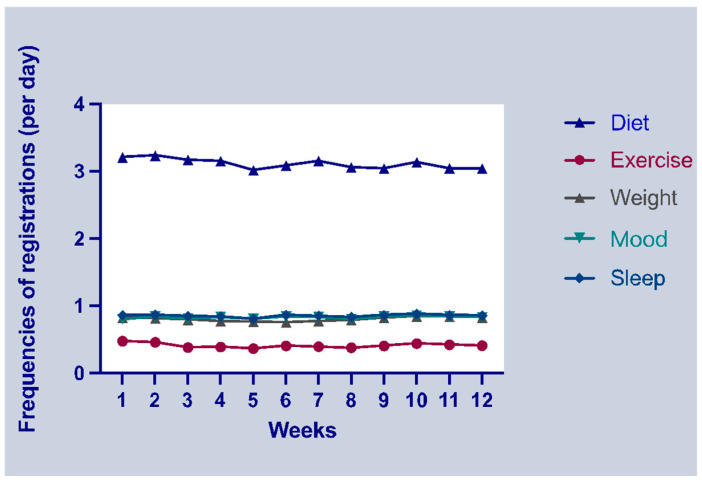
The mean frequencies of information registrations over three months.

**Figure 2 nutrients-16-02108-f002:**
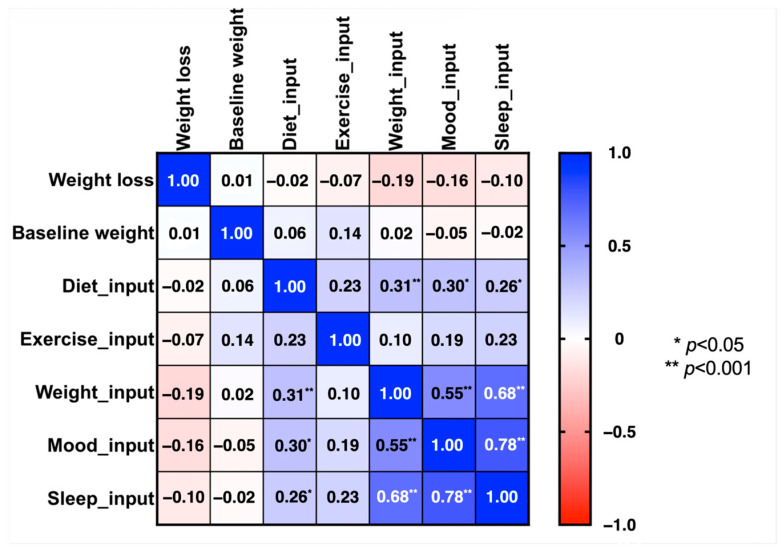
Spearman rank correlation matrix of app input frequency per day in the first week and weight loss over three months.

**Table 1 nutrients-16-02108-t001:** Baseline characteristics of participants (*n* = 68).

	Total	Women	Men
Participants, *n* (%)	68 (100.0)	18 (26.5)	50 (73.5)
Age (years), mean (SD)	42.3 (9.2)	40.8 (11.2)	42.8 (8.5)
BMI (kg/m^2^), mean (SD)	27.2 (3.2)	27.3 (3.3)	27.0 (3.1)
BMI 23–25, *n* (%)	16 (23.5)	0 (0.0)	16 (32.0)
BMI 25–30, *n* (%)	40 (58.8)	12 (66.7)	28 (56.0)
BMI ≥ 30, n (%)	6 (8.8)	6 (33.3)	0 (0.0)
Having exercise habit, *n* (%)	20 (29.4)	3 (16.7)	17 (34.0)
Having walking habit, *n* (%)	34 (50.0)	10 (55.6)	24 (48.0)
Faster walking speed, *n* (%)	41 (60.3)	8 (44.4)	33 (66.0)
Having medical history, *n* (%)	27 (39.7)	2 (11.1)	25 (50.0)
Having family medical history, *n* (%)	50 (73.5)	15 (83.3)	35 (70.0)
University graduate or higher, *n* (%)	49 (72.1)	6 (33.3)	43 (86.0)
Annual income of JPY 7 million or more, *n* (%)	40 (58.8)	7 (38.9)	33 (66.0)
Living alone, *n* (%)	14 (20.6)	5 (27.8)	9 (18.0)
Single, *n* (%)	21 (30.9)	11 (61.1)	10 (20.0)
Have lost more than 3 kg in the past, *n* (%)	34 (50.0)	10 (55.6)	24 (48.0)
Gained 10 kg more than weight at age 20, *n* (%)	4 (5.9)	1 (5.6)	3 (6.0)
Currently smoking, *n* (%)	5 (7.4)	0 (0)	5 (10.0)
Employed full-time, *n* (%)	61 (89.7)	12 (66.7)	49 (98.0)
Shiftwork, *n* (%)	2 (2.9)	0 (0)	2 (4.0)
Step count, mean (SD) ^#^	7998 (3585)	7739 (3368)	8223 (3518)

^#^ The step count values represent data for 67 participants (18 women and 49 men) owing to missing values from the accelerometer.

**Table 2 nutrients-16-02108-t002:** The median, minimum, and maximum values of input per day for each item for the first week.

Items	Mean (SD)	Median	Range
Diet input	3.21 (0.51)	3.29	(1.86, 4.00)
Exercise input	0.48 (0.52)	0.29	(0.00, 2.00)
Weight input	0.82 (0.26)	1.00	(0.14, 1.00)
Mood input	0.80 (0.27)	1.00	(0.00, 1.00)
Sleep input	0.86 (0.23)	1.00	(0.14, 1.00)

**Table 3 nutrients-16-02108-t003:** Predictors for 3% loss in initial weight in the univariate and multivariate logistic regression models.

Variables	Univariate Analysis		Multivariate Analysis *	
	OR (95%CI)	*p*-Value	OR (95%CI)	*p*-Value
Being a woman	0.395 (0.114, 1.370)	0.143		
Age	0.996 (0.944, 1.051)	0.891		
Have lost more than 3 kg in the past	1.895 (0.696, 5.157)	0.211		
Gained 10 kg more than weight at age 20	0.556 (0.055, 5.648)	0.619		
Having exercise habit	0.654 (0.214, 2.000)	0.457		
Having walking habit	0.308 (0.109, 0.870)	0.026 *	0.248 (0.079, 0.786)	0.018 *
Faster walking speed	0.258 (0.091, 0.731)	0.011 *	0.324 (0.105, 1.004)	0.051
Having medical history	2.244 (0.816, 6.168)	0.117		
Having family medical history	3.929 (1.009, 15.300)	0.049 *	4.269 (0.972, 18.746)	0.055
University graduate or higher	1.372 (0.445, 4.227)	0.581		
Annual income of JPY 7 million or more	1.407 (0.510, 3.882)	0.509		
Living alone	0.629 (0.174, 2.266)	0.478		
Single	0.589 (0.194, 1.792)	0.351		
Diet input in the first week	1.186 (0.446, 3.153)	0.732		
Exercise input in the first week	1.124 (0.439, 2.877)	0.807		
Weight input in the first week	10.644 (0.862, 131.375)	0.065		
Mood input in the first week	4.277 (0.523, 34.968)	0.175		
Sleep input in the first week	2.495 (0.244, 25.502)	0.441		

* Statistics of the regression model are as follows: omnibus tests of model coefficients, chi-square = 15.722, df = 3, *p* = 0.001; Cox & Snell R-square = 0.206, Nagelkerke R-square = 0.282; Hosmer–Lermeshow test: Chi-square = 8.474, df = 4, *p* = 0.076; model accuracy: 76.5%.

## Data Availability

The data presented in this study are available upon request from the corresponding author. The data are not publicly available because of privacy concerns.

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
