# Peer review of "Exploring Determinants of Successful Weight Loss with the Use of a Smartphone Healthcare Application: Secondary Analysis of a Randomized Clinical Trial"

_nutrients, 2024, doi:10.3390/nu16132108_

Round 1

Reviewer 1 Report

Comments and Suggestions for Authors

The authors conducted a post-hoc analysis of a previous RCT on the determinants of success with an eHealth weight loss intervention. Since it is to date quite difficult to achieve successful weight loss with a lifestyle intervention, and it has been shown hard to determine factors of success, improving our knowledge on this topic can be of importance.  

However, I have some concerns: 

  • 1. The duration of the intervention is only 3 months, so very short-term weight loss. Achieving long-term weight loss is a far more important problem to solve, but unfortunately this study does not address that. The authors mention this in their limitation section; however, this makes the study much less interesting for Nutrient readers. 

  • 2. The primary outcome on wich success was based (3% weight loss), was self-reported, leaving its reliability highly questionable. This is a serious flaw of this study, which cannot be controlled for. The authors also do not report on this in the study's limitations. 

  • 3. The result that a walking habit and faster walking pace were negatively associated with successful weight loss is somewhat strange. It is suggested that these people were able to improve more and therefore had greater weight loss. Since data was collected prospectively, the authors can analyse this: was there a greater improvement in movement seen in people who did not have a walking habit at baseline?

Comments on the Quality of English Language

no comments

Reviewer 2 Report

Comments and Suggestions for Authors

Comments to Authors

Introduction

·        “…. classified as obese”. à Always refer to people (or individuals) with overweight or with obesity, or as appropriate, but do not classify them as "obese" or similar. Revise all manuscript.

Methods

·        “(2) body mass index 23–40 kg/m2” à Why this range? If overweight starts at 25 kg/m2.

·        “(5) being deemed unsuitable for the study by the principal investigator for other reasons.” à for example?

Results

·        Table 1 à Add the number and percentage of participants according to overweight, obesity, etc.

Discussion

·        In the limitations section, I'm sure your study has more limitations, and you only mention two. One of them might be that people who enter data into the APP are considered to be adherent, but there is no guarantee that they will actually follow its recommendations. You can include others, I'm sure there are (and recognising the limitations is a strength of the study itself).

·        I also suggest that you include a paragraph before the limitations that explicitly lists the possible clinical or research implications of your study.
